# A dodecamethoxy[6]cycloparaphenylene consisting entirely of hydroquinone ethers: unveiling in-plane aromaticity through a rotaxane structure

Naoki Narita[1,4], Yusuke Kurita[1,4], Kohtaro Osakada[2], Tomohito Ide [3] ✉, Hidetoshi Kawai [1] ✉ & Yoshitaka Tsuchido [1] ✉

[*n*]Cycloparaphenylenes ([*n*]CPPs, where *n* is the number of phenylene groups), consisting of 1,4-linked phenylene unit, have attracted much attention due to their cyclic π-conjugated structures and physical properties. However, functionalizing of the benzene rings of smaller [*n*]CPPs (*n* < 7) has been a challenge due to ring strain and steric hindrance of the substituents that hampers their synthesis. Here we show successful synthesis of a new [6]CPP derivative with twelve methoxy groups at the 2,5-positions of all benzene rings by utilizing our developed CPP synthesis method via a macrocyclic gold complex. This molecule exhibited a significantly higher oxidation potential caused by the electron-donating ability of the methoxy groups and the tubular molecular conformation, allowing facile oxidation to give dicationic species with in-plane aromaticity. Furthermore, this molecule successfully included with the guest molecules with a flexible alkyl chain in the cavity, enabling the creation of a CPP-based rotaxane, which exploited its mechanically interlocked molecular structure to the first experimental observation that the in-plane aromaticity in the center of the macrocycle.

[*n*]Cycloparaphenylenes ([*n*]CPPs, where *n* is the number of phenylene groups), consisting of 1,4-linked phenylene unit, have attracted much attention due to their unique π-conjugated structures and physical properties[1–3]. Starting with the pioneering works in 2009–2010[4–6], [*n*] CPPs with various ring sizes and functional groups have been successfully synthesized over the past decade. Accordingly, tuning of physical properties and functionalization applications such as supramolecular hosts by introducing functional groups to the aryl ring have also been studied[7–9]. In particular, the introduction of well-ordered functional groups on all aromatic rings of [*n*]CPPs can lead to the occurrence of a unique molecular topology and physical properties due to the macrocyclic molecular structure with a periodical array, such as Tanaka's alkoxy-bridged [*n*]CPPs (*n* = 8, 10) with belt or Möbius topologies[10], Du's methoxy-functionalized [8]CPP[11], and Itami's fully-fluorinated [*n*]CPPs (*n* = 10, 12, 14, 16) (Fig. 1a)[12,13]. On the other hand, there are no examples of functionalization of all benzene rings of [*n*] CPPs with a small ring size, especially smaller than [7]CPP, due to the strong ring strain and the steric hindrance of the substituents that hampers their synthesis, although partially halogenated [5] and [6] CPPs have been synthesized by Yamago et al. (Fig. 1b)[14,15].

Recently, we have developed a new synthetic method for the [*n*] CPPs (*n* = 6, 9-15) via a macrocyclic gold complex[16,17]. A notable feature

[1]Department of Chemistry, Faculty of Science, Tokyo University of Science, 1–3 Kagurazaka, Shinjuku-ku, Tokyo 162-8601, Japan. [2]Laboratory for Chemistry and Life Science, Institute of Innovative Research, Tokyo Institute of Technology, 4259, Nagatsuta, Midori-ku, Yokohama 226-8503, Japan. [3]Department of Chemical Science and Engineering, National Institute of Technology, Tokyo College, 1220-2 Kunugida-machi, Hachioji-shi, Tokyo 193-0997, Japan. [4]These authors contributed equally: Naoki Narita, Yusuke Kurita. ✉e-mail: ide@tokyo-ct.ac.jp; kawaih@rs.tus.ac.jp; tsuchido@rs.tus.ac.jp

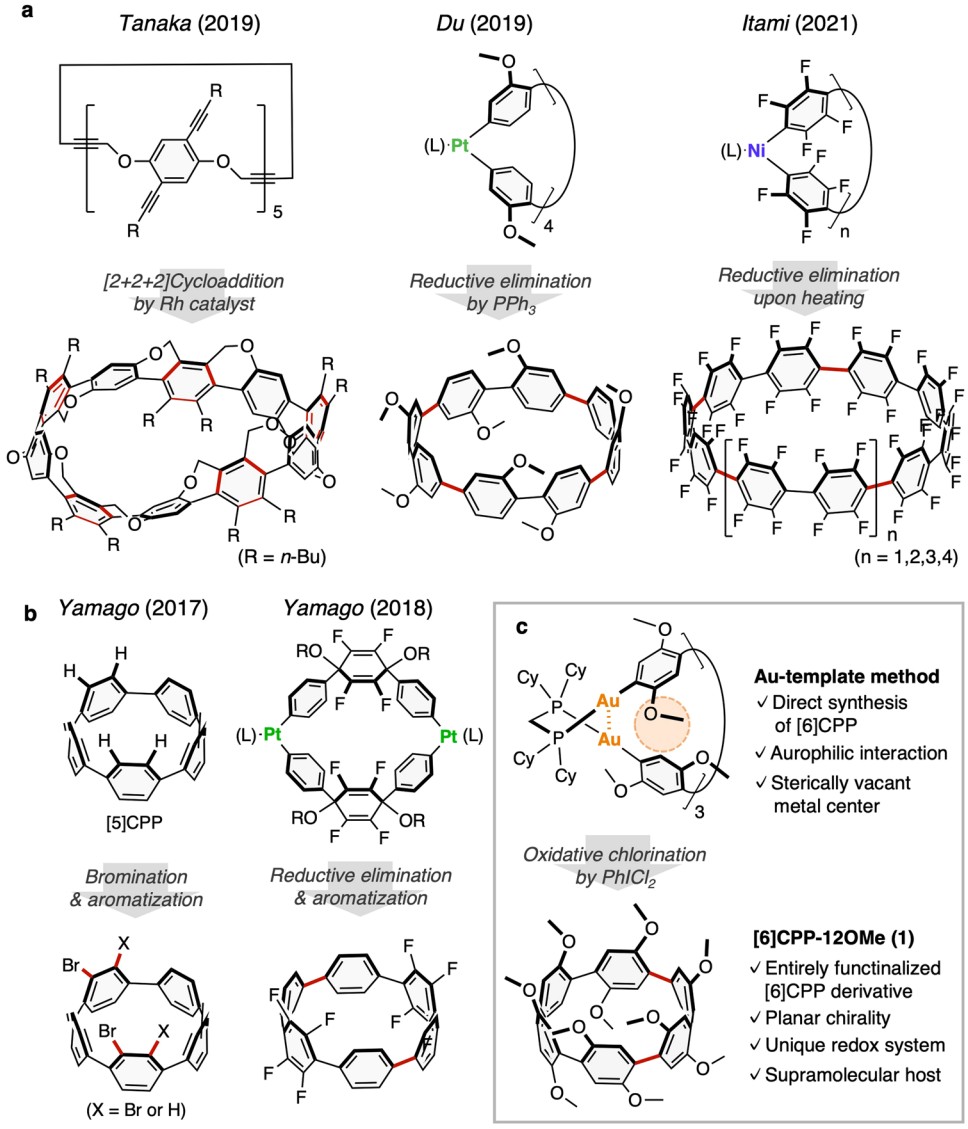

**Fig. 1 | Synthesis of [*n*]cycloparaphenylenes ([*n*]CPPs) with multiple functional groups. a** [*n*]CPPs (*n* ≥ 8) with substituents at all aromatic rings[10–13]. **b** Partially substituted [*n*]CPPs (*n* ≤ 6) with a small ring size[14,15]. **c** [6]Cycloparaphenylene derivatives consisting of hydroquinone ether units with 2,5-dimethoxy groups at all benzene rings, produced from macrocyclic Au complex (Cy = cyclohexyl) (This work)[16]. The red-colored bonds represent the bonds formed in the reaction.

of this method is that it employs a triangular-shaped macrocyclic gold complex with $Au_2$-diphosphine units located at the three corners that have stabilized their molecular structure through the aurophilic interactions[18–21]. We considered that our synthetic method would be suitable for the multi-functionalization of [*n*]CPPs with small ring size for the following reasons; i) the aurophilic interactions have degrees of freedom in their bond distances and angles, allowing the formation of the macrocyclic structure even when the multi-functionalization leads to crowding around the Au complex, ii) an Au(I) center coordinates only one aryl ligand, providing a more sterically vacant metal center than the biaryl M(II) complexes (M = Ni, Pd, Pt), and there would be less steric repulsion between the aryl ligands even if the substituents are introduced at the *ortho*-position (Fig. 1c).

Gaeta et al. reported the synthesis of a [8]CPP derivative incorporating one 2,5-dimethoxyphenylene ring and the inclusion ability toward methylpyridinium salts ($K_a = 2.2 \times 10^3 M^{-1}$, $CDCl_3$, 25 °C)[22]. Du et al. reported the synthesis of a [8]CPP derivative with several 2,5-dimethoxy groups[23], and that the molecule did not show significant inclusion ability toward various electron-deficient guest molecules. The averaged diameter of [6]CPP is 8.07 Å[24,25], which is comparable to

that of pillar[5]arenes (ca. 9.0 Å)[26], and is expected to recognize guest molecules with a size of alkyl chains that have not been achieved for the CPP system.

In this study, we report the synthesis of the [6]cycloparaphenylene derivative with hydroquinone ether units (2,5-dimethoxy groups) incorporated into all benzene rings by employing the Au-templated CPP synthesis method. In addition, we will explore the host-guest chemistry of this compound and its expansion into rotaxane synthesis. To prove an in-plane aromaticity[27–31] predicted in the dication species of cyclopayaphenylene, the electrochemical properties of these compounds will be evaluated.

## Results and discussion

### Synthesis and characterization of [6]CPP-12OMe

The reaction of 2,2′,5,5′-tetramethoxy-1,1′-biphenyl-4,4′-diyl diboronic acid (**L1**) with [$Au_2Cl_2$(dcpm)] (dcpm = bis(dicyclohexylphosphino) methane) in the presence of $Cs_2CO_3$ (1.0:1.0:6.0 molar ratio) in toluene/ethanol/water at 50 °C produced the macrocyclic Au complex, [$Au_2(C_6H_4$–2,5-$(OMe)_2)_2$(dcpm)]$_3$ (**Au-1**), in 84% isolated yield. Then, the oxidative chlorination[32] of **Au-1** occurs upon the addition of 3

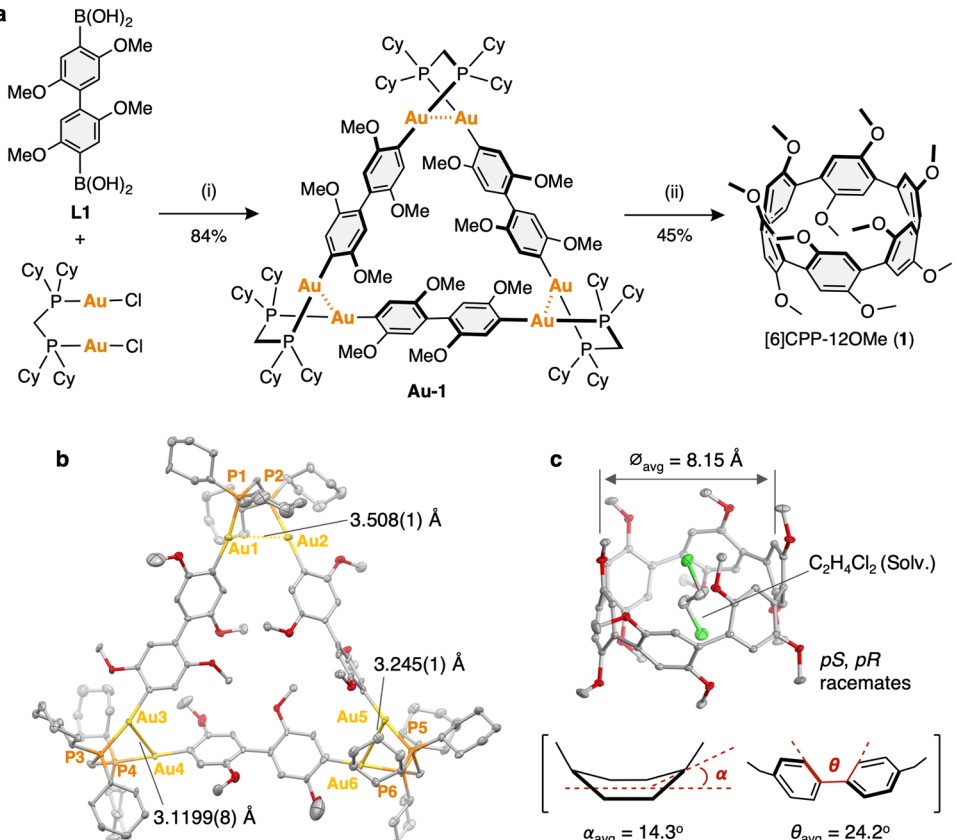

**Fig. 2 | Synthesis and characterization of [6]CPP-12OMe (1). a** Synthetic route for **1**. Reagents and conditions; (i) Cs$_2$CO$_3$ (6 equiv.), toluene/ethanol/water, 50 °C, 20 h, (ii) PhICl$_2$ (3 equiv.), DMF, −60 °C to 25 °C, 5.5 h (Cy = cyclohexyl). **b, c** The oak ridge thermal ellipsoid plot (ORTEP) drawings of (**b**) **Au-1** and (**c**) **1** (30% level of probability for thermal ellipsoids; gray, carbon; red, oxygen; yellow, gold; orange, phosphorus; light green, chlorine). Hydrogen atoms are omitted for clarity.

equivalents of PhICl$_2$ in DMF at −60 °C. The reaction mixture after stirring at 25 °C for 5 h was purified by silica gel column chromatography to give [6]CPP-12OMe (**1**) in 34% yield (Fig. 2a).

The molecular structure of macrocyclic Au complex (**Au-1**) was confirmed by X-ray crystallography (Fig. 2b). The complex adopted the triangular shaped structure with twisted Au-P-C-P-Au groups at the three corners, similar to that the non-substituted Au complex, [Au$_2$(C$_6$H$_4$)$_2$(dcpm)]$_3$, reported in our previous study (CCDC: 1994843)[16]. The distance between the two neighboring gold atoms in helical corners (3.508(1), 3.245(1), 3.1199(8) Å) are observed to be as significantly longer than that of [Au$_2$(C$_6$H$_4$)$_2$(dcpm)]$_3$ (3.1799(11), 3.0696(8), 3.0696(8) Å) (Supplementary Fig. 31), which indicates that the triangular structure can be maintained even when the bulky methoxy groups are introduced at the *ortho* positions of the arylene linkers. This result clearly showed that our Au-templated method has a wide range of substrate applicability.

The single crystals of **1** suitable for X-ray crystallography were obtained via the vapor diffusion of *n*-hexane into 1,2-dichloroethane solution of the molecule. The crystal structure was analyzed as 1,2-dichloroethane incorporated structure (Fig. 2c). The all-methoxy groups of the dimethoxybenzenes are oriented in the same direction in the molecule, resulting in two enantiomers, *pR* and *pS*, with planar chirality depending on the orientation of the methoxy groups. The molecule adopted an elliptical molecular structure similar to [6]CPP with a slightly lengthened diameter (⌀$_{avg}$ = 8.15 Å). Interestingly, the averaged dihedral angle between two phenylene rings ($\theta_{avg}$) was estimated to be 24.2°, which is narrower than that of [6]CPP (28.1°)[24,25]. Rathore reported synthesis, molecular structure, and electrochemical properties of oligo(*p*-phenylene)s with 2,5-dimethoxy groups, (C$_6$H$_4$−2,5-(OMe)$_2$)$_n$H$_2$ (*n* = 2-9)[33], which are a linear analog of **1**. The

dihedral angle (56°) in the crystal structure of (C$_6$H$_4$−2,5-(OMe)$_2$)$_3$H$_2$ was significantly wider than that of *p*-terphenyl (≤1°)[34], which is in contrast to that of the CPP system in this study.

The strain energy of **1**, as simulated through the density functional theory (DFT) calculations (ωB97M-V/def2-TZVP//TPSS-D3(BJ)/def2-SVP) based on the homodesmotic reaction using ORCA[35] (Supplementary Table 3), was unexpectedly estimated to be lower (88.2 kcal mol⁻¹) compared to that of [6]CPP (92.6 kcal mol⁻¹) calculated at the same level, despite the introduction of bulky methoxy groups. Further computational studies, including bond critical points (BCPs) / quantum theory of atoms in molecules (QTAIM) and non-covalent interaction (NCI) analyses using Multiwfn[36] and NCIPLOT[37] programs, have demonstrated the formation of intramolecular CH...O hydrogen bonds between methoxy oxygens and aryl hydrogens of another benzene ring (Supplementary Figs. 18-19). Thus, the lower strain energy of **1** compared to [6]CPP can be attributed to a contribution from stabilization through the multiple hydrogen bonds. This interaction tends to be weak in a less-strained molecule such as (C$_6$H$_4$−2,5-(OMe)$_2$)$_3$H$_2$[33] by a large Ar-Ar dihedral angle primarily due to the steric hindrance. Therefore, the formation of the tubular conformation in **1** can be ascribed to the restricted change in dihedral angles due to the strained macrocyclic structure, along with the structure-specific intramolecular CH...O hydrogen bonds.

The above-mentioned molecular conformations in the crystal structures have also been observed in solution. The ¹H NMR signal of arylene protons (Ar-H) of **1** appeared as a slightly broad singlet peak at 7.34 ppm (400 MHz, CDCl$_3$, 298 K). The fact that the corresponding signal of (C$_6$H$_4$−2,5-(OMe)$_2$)$_n$H$_2$[33] was observed in 6.9 ppm suggests that all arylene rings of **1** are oriented perpendicular to the CPP macroring as in the crystal structure, and thus the downfield shift of the

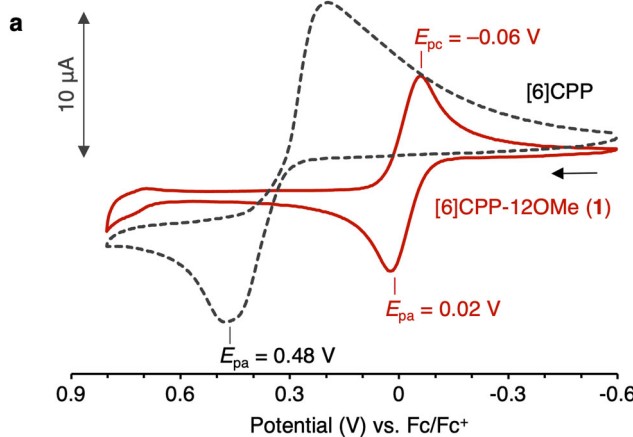

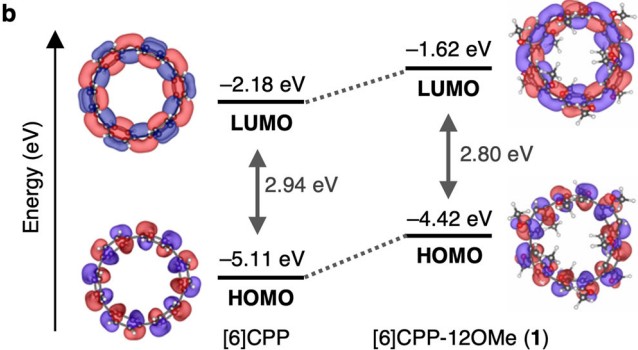

**Fig. 3 | Electrochemical properties of [6]CPP-12OMe (1). a** Cyclic voltammograms of **2** (red solid line) and [6]CPP (black dashed line) (CH$_2$Cl$_2$, [M] = 1.0 × 10$^{-3}$ M, electrolyte: 0.1 M TBAPF$_6$, scan rate: 100 mV s$^{-1}$, r.t.) ($E_{pa}$: anodic peak potential, $E_{pc}$: cathodic peak potential). **b** Energy diagrams and Kohn-Sham orbitals of **1** (right) and [6]CPP (left) (B3LYP/def2-SVP//TPSS-D3(BJ)/def2-SVP, isosurface; 0.02). Source data are available as a Source data file.

Ar-H proton of **1** compared to that of (C$_6$H$_4$−2,5-(OMe)$_2$)$_n$H$_2$ would be attributed to less shielding effects from the other arylene rings due to the small Ar-Ar dihedral angles. The variable-temperature NMR measurements, DFT calculations, and chiral high performance liquid chromatography (HPLC) analysis suggested that the ring flipping of **1** occurred rapidly, and that the two enantiomers are fast exchanging in solution (Supplementary Figs. 14–16).

The cyclic voltammogram of **1** showed the reversible redox wave with $E_{1/2}$ = 0.004 V vs. Fc/Fc$^+$ ($E_{pa}$ = 0.02 V, $E_{pc}$ = −0.06 V), which is drastically lower potential than that of [6]CPP ($E_{pa}$ = 0.48 V)[24,25] (Fig. 3a). To understanding the difference in the electrochemical properties, the frontier orbitals were simulated by DFT calculations[35] (Fig. 3b). Interestingly, although the shape of the frontier orbitals was comparable, the highest occupied molecular orbital (HOMO) and the lowest unoccupied molecular orbital (LUMO) levels of **1** (−4.42, −1.62 eV) were found to be drastically higher than those of [6]CPP (−5.11, −2.18 eV). The significant increase in these levels would be due to i) the effect of the electron-donating nature of the methoxy groups and ii) the increase in ring strain and π-conjugation based on the decrease in Ar-Ar dihedral angle caused by the intramolecular CH…O hydrogen bonds of the methoxy groups[38].

## Chemical Oxidation of [6]CPP-12OMe

The chemical oxidation of [6]CPP-12OMe (**1**) with [N(C$_6$H$_4$−4-Br)$_3$]SbCl$_6$ (Magic Blue, **MB**) was monitored by ultraviolet/visible/near-infrared (UV/vis/NIR) spectra (Fig. 4a). In the neutral state, compound **1** showed absorptions at 320 and 390 nm in the UV region and a broad absorption around 450-600 nm, resulting in a reddish-brown color.

Upon additions of 0.5 and 1.0 equiv. of **MB**, a broad absorption over the near-infrared region was observed ($\lambda_{max}$ = 1400 nm), turning a green solution. Further addition of **MB** up to 2.0 equiv. decreased the absorption around 1400 nm and absorptions at 615 and 970 nm appeared, and the solution turned blue (Fig. 4b). These results are similar to those reported by Yamago et al.[27–31] about the formation of cation radicals ([6]CPP$^{•+}$) ($\lambda_{max}$ = 1136 nm) and dication ([6]CPP$^{2+}$) ($\lambda_{max}$ = 464, 792 nm) in the multi-step oxidation of [6]CPP by (NO) SbF$_6$ (nitrosonium hexafluoroantimonate). The time-dependent density functional theory (TD-DFT) calculations suggested the formation of radical cation (**1**$^{•+}$) ($\lambda_{DFT}$ = 1334 nm) and dication (**1**$^{2+}$) ($\lambda_{DFT}$ = 971 nm), respectively. The electron density difference of the first excited states in both **1**$^{•+}$ and **1**$^{2+}$ from those of [6]CPP$^{•+}$ and [6]CPP$^{2+}$ suggests an electron donation from the methoxy groups regarding the red shifts of the absorption bands (Supplementary Tables 4–6).

The formation of dication (**1**$^{2+}$) was also confirmed by $^1$H NMR spectroscopy. The chemical shift of the arylene protons was shifted from 7.34 ppm to 5.74 ppm by addition of 2 equiv. of **MB** (CDCl$_3$, 298 K, Supplementary Fig. 20), which is due to the shielding effect caused by a generation of an in-plane aromaticity[28]. The results of the nucleus-independent chemical shifts (NICS) calculation also support the in-plane aromaticity of the oxidized species (Fig. 4c). The NICS value at the center of the macroring was strongly negative (−16.1 ppm), suggesting a strong diatropic ring current induced throughout the entire cavity due to the formation of in-plane aromaticity (Fig. 4d). Furthermore, negative NICS values over 10 Å along the perpendicular directions (z-axis) from the macroring suggest that the induced magnetic field extends also in the z-axis direction due to the ring current on the macroring (Fig. 4e).

A single crystal of the **1**$^{2+}$ was obtained by vapor diffusion (1,2-dichloroethane/n-hexane), and the structure was successfully analyzed by X-ray crystallography (Fig. 4f). This is the smallest ring-sized CPP dication that has been successfully determined by X-ray crystallography, although Yamago et al. reported the crystal structure of [8] CPP dication[27]. The harmonic oscillator model of aromaticity (HOMA) value of each 6-membered ring in **1**$^{2+}$ was calculated to be 0.585, which was a significant decrease of more than 0.3 over those of the neutral species **1** (0.895), indicating that the benzene rings in the dication adopt a quinoid-like structure resulting in the loss of aromaticity of the benzene ring. The bond alternation analysis of **1** and **1**$^{2+}$ revealed that the arylene units of **1**$^{2+}$ prefers a quinoid-like structure rather than quinone-like structure. According to the study on the oxidation of (C$_6$H$_4$−2,5-(OMe)$_2$)$_n$H$_2$ (n = 2-9) by Rathore et al.[33], the linear oligomers favored the quinone-like structure with the large dihedral angle, and the positive charge being localized to a part of the arylene ring(s), which are in contrast with the results of **1**$^{2+}$.

## Host-guest chemistry of [6]CPP-12OMe

[6]CPP-12OMe (**1**) adopts the cylindrical molecular structure with extremely electron-rich aryl rings owing to the multi-substituted methoxy groups, promising applications in host-guest chemistry. First, to investigate the scope of guest applicability for **1**, we have mixed **1** and various guest molecules in CDCl$_3$ and found that the chemical shift of the Ar-H protons changed significantly toward n-alkanes with an electron-deficient group such as cyano, N-pyridinium, azide, and iso-cyanate groups. On the other hand, mixing with guest molecules with branched chains (e.g. cyanoalkanes with branched chains, and trialkylammonium salts) and with electron-rich molecules (e.g., triglyme) did not affect the chemical shift of the Ar-H protons (Supplementary Figs. 24–26).

The complexation of **1** with NC-(CH$_2$)$_4$-CN (adiponitrile, **G1**) was confirmed by $^1$H NMR titrations (Fig. 5a). The addition of **G1** upon a CDCl$_3$ solution of **1** (0.5 mmol L$^{-1}$, 298 K) caused a downfield shift of the $^1$H NMR signals of aryl and methoxy protons (Fig. 5b), indicating the formation of a supramolecular complex with a faster equilibrium

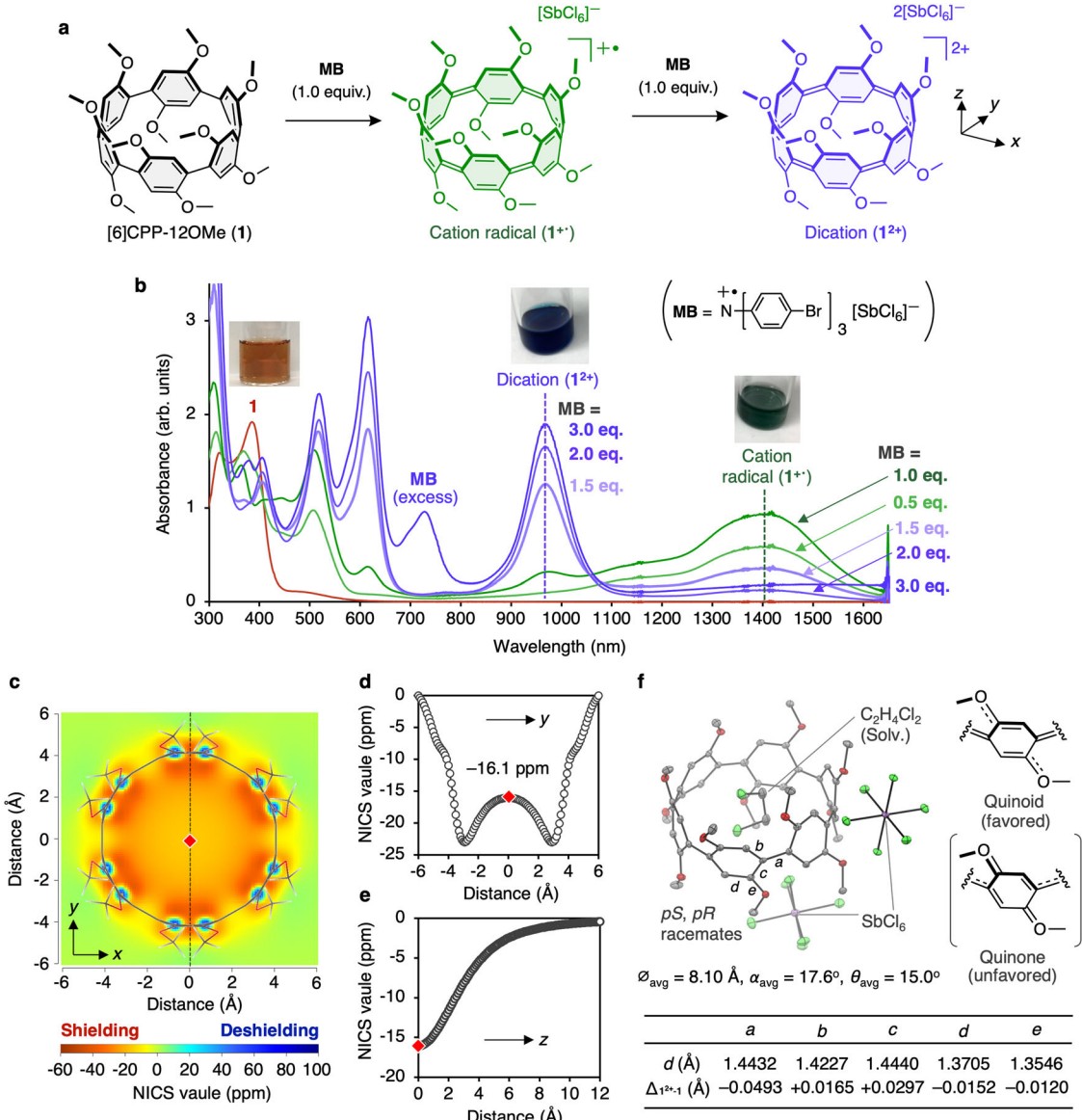

**Fig. 4 | Chemical oxidation of [6]CPP-12OMe (1) with [N(C₆H₄−4-Br)₃]SbCl₆ (Magic Blue, MB).** a Reaction scheme. b Ultraviolet/visible/near-infrared (UV/vis/NIR) titration of **1** with **MB** (CH₂Cl₂, [**1**] = 1.5 × 10⁻⁵ M, r.t.). c Two-dimensional nucleus-independent chemical shifts (2D NICS) grid of dication (**1²⁺**). d NICS scan relative to *y*-axis. e NICS scan values relative to *z*-axis. f ORTEP drawing of **1²⁺** (30% level of probability for thermal ellipsoids; gray, carbon; red, oxygen; light green, chlorine; antimony, purple). Hydrogen atoms are omitted for clarity. Source data are available as a Source data file.

rate than the ¹H NMR timescale. The association constant ($K_a$) was estimated to be 186 M⁻¹ (CDCl₃, 298 K)[39]. The change in chemical shift would be caused by deshielding of protons due to a decrease in the dihedral angles ($\theta$) accompanying a guest inclusion, rather than by an electrostatic interaction between the host and guest molecules.

Use of symmetry adaptive perturbation theory (SAPT) calculations[40] with PSI4[41] to decompose the interaction energies for different conformations and positions of **G1** in the inclusion complex (**1⊃G1**) showed that having **G1** located inside the cavity with the *gauche*-conformation brings the CH₂ unit closer to the arylene unit, which affects the electrostatic and dispersion forces more effectively than the *anti*-conformation (Fig. 5c). In addition, the *gauche*-conformation also enables a stronger electrostatic interaction between the terminal cyano group and electron-rich skeleton of **1** because of closer to each other than the *anti*-conformation. The closer guest-host contact enforces the dispersion force, which is in turn offset (to some extent) by the increased exchange repulsion.

A single crystal of the inclusion complex (**1⊃G1**) suitable for single crystal X-ray crystallography was obtained by slow diffusion of *n*-hexane vapor into a CHCl₃ solution of **1** with an excess amount of **G1** (Fig. 5d). The two enantiomers of **1** (*pR* and *pS*) and four **G1** molecules are in an unit cell (*P2₁/c*). Two **G1** molecules are located inside of the cavity of two **1** molecules with a *gauche*-conformation, whereas another two **G1** molecules are located outside of the macrocycle with an *anti*-conformation. The averaged dihedral angle ($\theta_{avg}$) of inclusion complex (**1⊃G1**) was estimated to be 9.86°, which is drastically reduced from that of 1,2-dichloroethane incorporated structure (24.2°) shown in Fig. 2c. The guest molecule inside the cavity of **1** adopts an entropically unfavorable the *gauche*-conformation, and multiple intermolecular CH-π interactions were formed between the alkyl hydrogens of **G1** and the arylene rings of **1** (Supplementary Fig. 34c). These results suggest that the inclusion of **G1** is enthalpy-driven process caused by the combination of dispersion and electrostatic interactions between host and guest molecules, which has been supported

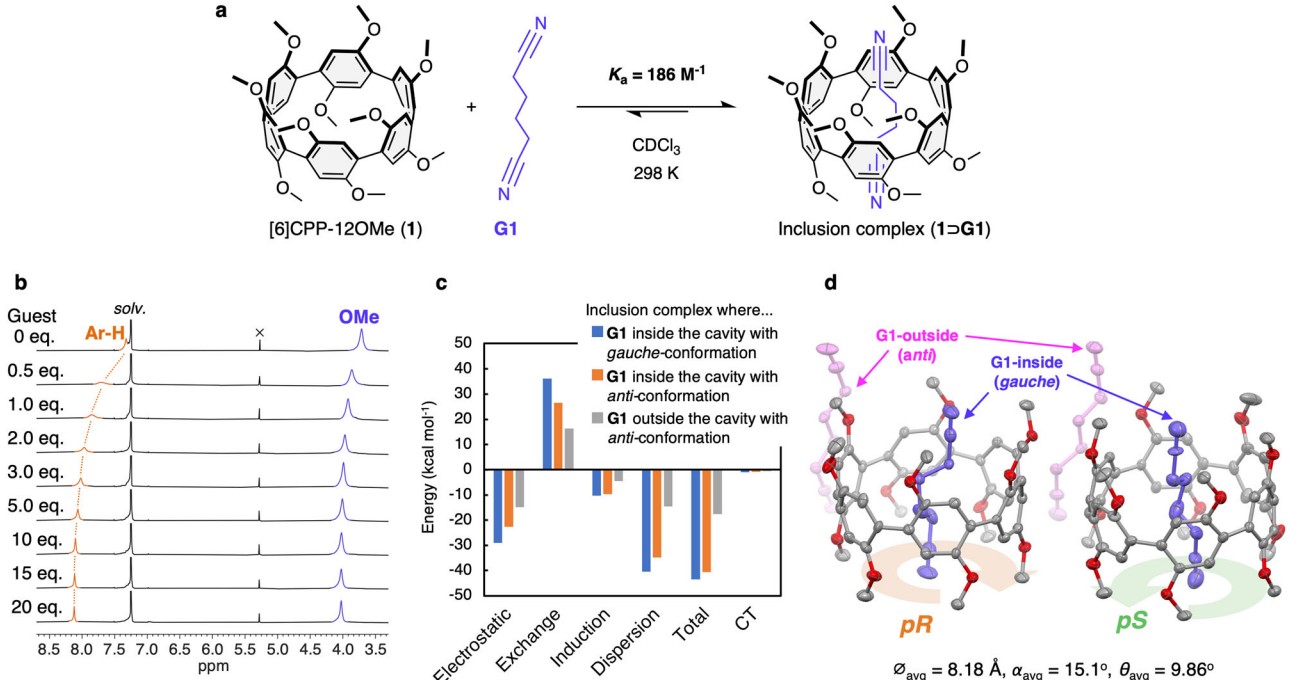

**Fig. 5 | Host-guest chemistry of [6]CPP-12OMe (1). a** Formation of inclusion complex (**1⊃G1**) between **1** and NC-(CH₂)₄-CN (adiponitrile, **G1**). **b** ¹H NMR titration of **1** with **G1** (400 MHz, CDCl₃, r.t.). **c** Interaction energy decomposition by symmetry-adapted perturbation theory (SAPT) calculation of **1⊃G1** with different positions and orientations of **G1** relative to **1** (SAPT0/jun-cc-pVDZ level of theory). **d** ORTEP drawing of inclusion complex (**1⊃G1**) (30% level of probability for thermal ellipsoids; gray, carbon; red, oxygen; **G1** inside of the cavity of **1**, blue; **G1** inside of the cavity of **1**, pink). Hydrogen atoms are omitted for clarity.

by the results of the SAPT calculations (Fig. 5c). This is the first example of CPP encapsulating a guest molecule with a flexible alkyl structure, which is achieved by the electron-rich arylene rings with the methoxy groups.

## Rotaxane synthesis and its chemical oxidation

We then applied the function of [6]CPP-12OMe (**1**) as a supramolecular host to the synthesis of a mechanically interlocked molecule (MIM)[42] with a planar chirality. Mixing **1** with an excess amount of OCN-(CH₂)₆-NCO (hexamethylene diisocyanate) in CHCl₃ to form a pseudo[2] rotaxane in situ, then introducing a bulky substituent at both ends of the axle molecule through a urea bond formation reaction with 1-adamantanemethylamine produced a [2]rotaxane (**2**) containing **1** as a ring component (Fig. 6a). The chiral HPLC analysis of **2** observed two peaks with the retention times of 12.5 and 13.2 min, respectively (Supplementary Fig. 14b). This result indicates that the interpenetration of the alkyl chain to the cavity of **1** prevented the *pR* ⇌ *pS* racemization through a sequence of "oxygen-through-the-annulus" inside-out and outside-in rotations as seen in pillar[*n*]arene[43].

In the ¹H NMR spectrum of racemate of **2** (Fig. 6b, iii), the chemical shifts of the CH₂ protons of the axle component (H_c, H_d) were observed around 0.9 to −0.5 ppm, which are upfield region than that of axle compound (**A**) (Fig. 6b, ii). These shifts are attributed to the shielding effect from the arylene rings of **1**, indicating that the alkyl chain is mechanically interlocked inside the cavity of **1**. On the other hand, one of the NH protons of the urea groups shifted significantly to a downfield position at 5.79 ppm, which is due to the formation of NH...O hydrogen bonds between NH proton and methoxy oxygens of **1**. The formation of NH...O hydrogen bonds was also observed in the stable conformation of **2** as estimated by DFT calculations (Fig. 6a). These results suggest that the terminal urea groups function as a "station", and that the CPP macrocycle undergoes a rapid shuttling movement along the alkyl chain of the axle component, which is also supported by the molecular dynamics (MD) calculations (Fig. 6c and

Supplementary Movie 1). von Delius et al. reported the synthesis of [2] rotaxane with [10]CPP as the ring component and fullerene (C₆₀) as the axle and the endcap groups[44]. In that case, the CPP macrocycle mainly rotates around C₆₀ moiety because of the strong interaction between [10]CPP and C₆₀. Recently, Jasti et al. successfully synthesized a [3] rotaxane with two N-doped [8]CPPs as the ring component and an axle component consisting of 1,4-phenylene and ethynylene groups, utilizing an active template method[45]. As there are no stations incorporated in the axle component, the CPP units are assumed to be shuttling freely along the axial molecule. Thus, we succeeded in developing a new CPP-based molecular machine that imparts a higher order of mechanical motion by utilizing the interaction between methoxy groups introduced into [6]CPP and the urea moieties of the axle component. This molecule offers the potential for a new supramolecular system utilizing its planar chiral mechanically interlocked structure, redox ability of CPP unit, and an anion-binding property of urea moieties[46].

We have successfully achieved the selective two-electron oxidation of the ring component within the [2]rotaxane (**2**), without decomposition of other component, by adding 2 equivalents of **MB** as a mild oxidant owing to the low oxidation potential of [6]CPP-12OMe (**1**) (Fig. 7a). Remarkably, two-electron oxidation of **2** causes significant upfield shift of the ¹H NMR signals of the methylene protons of the axle molecule to −4.6 (H_b), −8.9 (H_c), and −14.5 (H_d) ppm, respectively (Fig. 7b). There significant upfield shifts can be ascribed to the shielding effect by a diatropic ring current on the CPP macroring, leading to the appearance of the above-mentioned in-plane aromaticity (Fig. 4c). In previous studies on CPP dications, the presence of in-plane aromaticity has been indirectly speculated from the chemical shifts of the aryl protons located in the perpendicular position of the macrorings[27–30]. However, it was not clear whether these changes in chemical shifts were due to the diatropic ring current caused by the in-plane aromaticity or the formation of the cyclic quinoid structures. Anderson et al. have demonstrated the in-plane

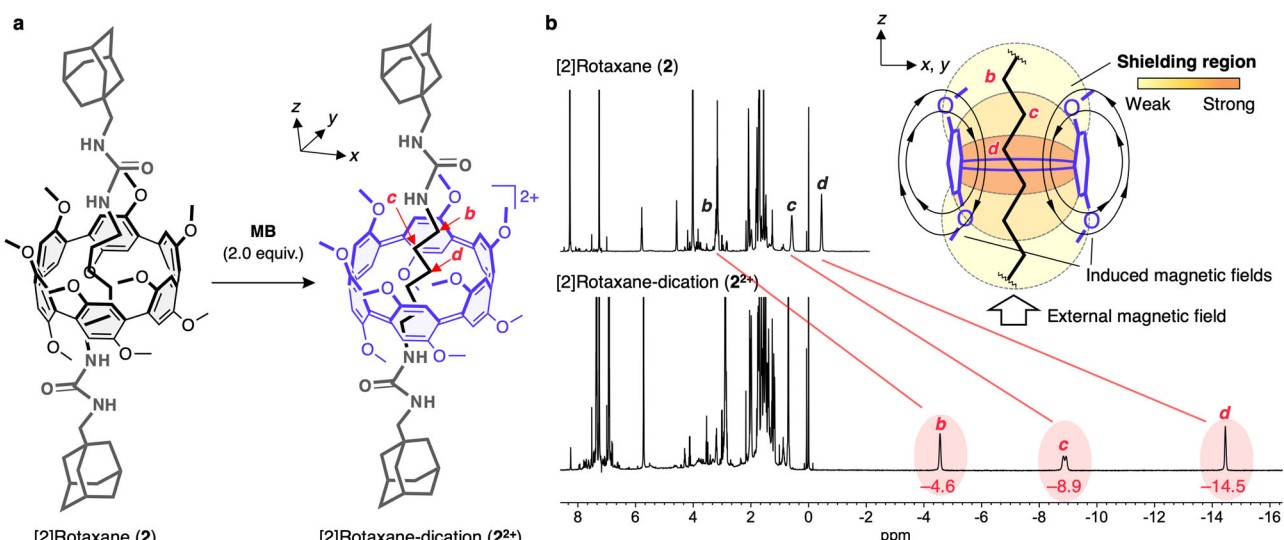

**Fig. 6 | Synthesis and characterization of [2]rotaxane (2). a** Synthetic route for **2**. Molecular model of **2** was optimized by DFT calculations. Hydrogen atoms of CPP unit were omitted for clarity. **b** ¹H NMR spectra (400 MHz, CDCl₃, r.t.) of (i) [6]CPP-12OMe (**1**), (ii) axle compound (**A**) (the molecular structure is shown in the inset), and (iii) [2]rotaxane (**2**). **c** The distance change between the centroid of CPP macroring and outer nitrogen atom in the axle as a function of time during the molecular dynamics (MD) simulations. Source data are available as a Source data file.

**Fig. 7 | Chemical oxidation of [2]rotaxane (2). a** Reaction scheme. **b** ¹H NMR spectra (400 MHz, CDCl₃, r.t.) of [2]rotaxane (**2**) before (top) and after (bottom) chemical oxidation upon addition of 2 equivalents of Magic Blue (**MB**). Inset: plausible illustration of the shielding area of the in-plane aromaticity based on the chemical shifts of the alkyl chain.

aromatic/anti-aromatic nature of multiply oxidized porphyrin nanorings, as evidenced by the chemical shifts of pyridyl hydrogen in hexa(4-pyridylethynyl)benzene, which was incorporated within the nanoring's interior[47,48]. However, the environment at the center of the nanoring could not be evaluated for the chemical shifts of the pyridyl hydrogens, as they are positioned in close proximity to the nanoring by coordinating to the porphyrin groups. In this study, we present the first experimental evidence of the in-plane aromaticity in the center of the macrocycle, as indicated by the chemical shifts of protons of the interlocked alkyl chain threaded in the perpendicular direction to the CPP macrocycle. Furthermore, the sequential downfield shift in the $^1$H NMR signals of the methylene protons from the central position along the urea side ($H_d$: −14.5 ppm → $H_c$: −8.9 ppm → $H_b$: −4.6 ppm) provides experimental evidence for decreasing shielding of the in-plane aromaticity along the perpendicular ($z$-axis) direction to the CPP macroring (Fig. 7b, inset).

## Methods

### General

All manipulations were carried out under an argon atmosphere using standard Schlenk techniques. The $^1$H, $^{13}$C{$^1$H}, and $^{31}$P{$^1$H} NMR spectra were recorded on a Bruker Biospin AVANCE NEO 400 (400 MHz), Bruker Biospin AVANCE II DPX-400 (400 MHz), Bruker Ascend™ 400 (400 MHz), or JEOL ECZ400S (400 MHz). The chemical shifts in $^1$H and $^{13}$C{$^1$H} NMR were referenced with respect to TMS ($^1$H NMR; $\delta$ 0.00 for CDCl$_3$) or residual peaks of the solvents used ($^{13}$C NMR; $\delta$ 77.16 for CDCl$_3$). The peak positions of the $^{31}$P{$^1$H} NMR spectra were referenced to external 85% H$_3$PO$_4$ ($\delta$ 0.00) in deuterated solvents). Column chromatography was performed on silica gel 60 (spherical, neutral) with particle size of 40-63 μm (YMC), or 40-100 μm (Kanto), or 100-210 μm (Kanto). Deuterated solvents were purchased from Kanto or ISOTEC and used as received. Chiral HPLC analysis were conducted on an instrument equipped with a JASCO PU-4185 HPLC pump, JASCO UV4575 UV/Vis detector, and a DAICEL CHIRALPAK® IA column (10 mm × 250 mm). Elemental analyses were performed using a J-Science JM10. The high-resolution mass spectrometry (HRMS) was measured in Spiral mode (positive) on a JEOL JMS-S3000 SpiralTOF (MALDI-TOF). Cyclic voltammetry (CV) was measured in CH$_2$Cl$_2$ solutions containing 100 mM $n$Bu$_4$NPF$_6$ with ALS electrochemical analyser Model-600A. The measurement was carried out in a standard one compartment cell equipped with a Ag/Ag$^+$ reference electrode, a platinum wire counter electrode, and a platinum disk working electrode. UV-Vis-NIR absorption spectra were obtained on a SHIMADZU UV-3150PC spectrophotometer.

The commercially available reagents, HAuCl$_4$·4H$_2$O (Tanaka, purity: >99.8%), tetrahydrothiophene (TCI, purity: >99.0%), bis(dicyclohexylphosphino)methane (Wako, purity: 95 + %), 1-bromo−2,5-dimethoxybenzene (Aldrich, purity: 98%), 2,5-dimethoxyphenylboronic acid (Wako, purity: >97 + %), NBS (Kanto, purity: >98.0%), $n$-BuLi in 1.6 M $n$-hexane solution (Kanto, purity: >98.0%), B($i$-OPr)$_3$ (Aldrich, purity: ≥98%), Cs$_2$CO$_3$ (Wako, purity: 95.0 - 102.0%), Magic Blue (MERCK, technical grade), hexamethylene diisocyanate (TCI, purity: >98.0%), 1-adamantanemethylamine (TCI, purity: >98.0%), were used as received. [Pd(PPh$_3$)$_4$] was prepared by hydrazine reduction of a mixture of PdCl$_2$ and PPh$_3$. PhICl$_2$ was prepared from the oxidation of iodobenzene by Cl$_2$, which was generated in situ from the reaction of 5% NaClO aq. (commercial household bleach) with conc. HCl. [Au$_2$Cl$_2$(dcpm)] (dcpm = bis(dicyclohexylphosphino)methane) was prepared by the ligand exchange reaction of [AuCl(tht)] (tht = tetrahydrothiophene) with 0.5 molar amount of dcpm (Supplementary Method 1). (2,2′,5,5′-Tetramethoxy-[1,1′-biphenyl]−4,4′-diyl)diboronic acid (**L1**) was synthesized by the Suzuki coupling reaction of 1-bromo−2,5-dimethoxybenzene with 2,5-dimethoxyphenylboronic acid, followed by bromination and borylation at the 4,4′-position (Supplementary Methods 2–4).

### Synthesis of macrocyclic Au complex (Au-1)

A mixture of **L1** (0.87 g, 2.4 mmol), Cs$_2$CO$_3$ (4.71 g, 14.4 mmol) and [Au$_2$Cl$_2$(dcpm)] (2.11 g, 2.4 mmol) in degassed toluene/ethanol/water (100 mL/25 mL/25 mL) was stirred for 20 hours at 50 °C under argon atmosphere. After the reaction mixture was allowed to cool to room temperature, the precipitate was collected by suction filtration. The crude product was purified by recrystallization (CHCl$_3$/$n$-hexane = 10 mL/40 mL) to afford **Au-1** as a white solid (2.15 g, 0.67 mmol, 84%).

$^1$H NMR (400 MHz, CDCl$_3$, 297 K): $\delta$ 7.17 (t, $J$ = 3.1 Hz, 6H, Ar-H), 6.75 (s, 6H, Ar-H), 3.70 (s, 18H, OCH$_3$), 3.31 (s, 18H, OCH$_3$), 2.29-2.18 (br, 12H, C$_6$H$_{11}$, CH$_2$), 2.14-2.07 (br, 24H, C$_6$H$_{11}$), 2.07-1.97 (br, 12H, C$_6$H$_{11}$), 1.92-1.85 (br, 24H, C$_6$H$_{11}$), 1.78-1.69 (br, 18H, C$_6$H$_{11}$), 1.69-1.63 (br, 12H, C$_6$H$_{11}$), 1.30-1.25 (br, 36H, C$_6$H$_{11}$).

$^{13}$C{$^1$H} NMR (100 MHz, CDCl$_3$, 297 K): $\delta$ 161.58 (t, $J$ = 54.9 Hz), 152.23 (s), 126.69 (s), 124.77 (s), 115.74 (s), 57.97 (s), 56.60 (s), 35.45 (t, $J$ = 14.2 Hz), 29.92 (s), 29.16 (s), 26.98 (t, $J$ = 6.96 Hz), 26.14 (s).

$^{31}$P{$^1$H} NMR (161 MHz, CDCl$_3$, 297 K): $\delta$ 48.44 (s)

Anal. Calcd for C$_{123}$H$_{186}$O$_{12}$Au$_6$P$_6$ + CHCl$_3$: C, 44.54; H, 5.64. Found: C, 44.93; H, 5.81.

### Synthesis of [6]CPP-12OMe (1)

To a suspension of **Au-1** (97.4 mg, 30 μmol) in degassed DMF (20 mL) was added PhICl$_2$ (26.4 mg, 96 μmol) in DMF (10 mL) dropwise with stirring at −60 °C for 30 min under an argon atmosphere in the dark. The reaction mixture was stirred at the same temperature for 30 min, then it was allowed to warm to 25 °C and stirred for 5 h in the dark. Solvent and iodobenzene (by-product) were removed under vacuum. The crude product was purified by silica gel column chromatography in the dark (eluent; CHCl$_3$/AcOEt = 10:1, $R_f$ = 0.6) gave **1** (11.0 mg, 13.5 μmol, 45%) as an orange solid.

$^1$H NMR (400 MHz, CDCl$_3$, 297 K): $\delta$ 7.33 (s, 12H, Ar-H), 3.71 (s, 36H, OCH$_3$)

$^{13}$C{$^1$H} NMR (100 MHz, CDCl$_3$, 297 K): $\delta$ 152.39 (s), 126.21 (s), 115.65 (s), 57.22 (s)

HRMS (MALDI-TOF, DCTB; Ionizer, NaTFA): calcd. for C$_{48}$H$_{48}$O$_{12}$: 816.3140, found: $m/z$ 816.3159 ([M]$^+$).

### Synthesis of [2]rotaxane (2)

To a CHCl$_3$ solution (3 mL) of [6]CPP-12OMe (**1**) (51.8 mg, 63 μmol) and hexamethylene diisocyanate (529 mg, 3.1 mmol) was added 1-adamantanemethylamine (1.19 g, 7.2 mmol) at 0 °C, and stirred for 10 min at the same temperature in the dark. After the reaction mixture allowed to warm to r.t., CHCl$_3$ (6 mL) was added the reaction mixture. The resulting white solids was removed by suction filtration. The filtrate was purified by silica gel column chromatography twice in the dark (eluent; CHCl$_3$/AcOEt = 10:3, $R_f$ = 0.24) to give [2]rotaxane (**2**) (2.1 mg, 1.6 μmol, 3.0%) as a dark orange solid.

$^1$H NMR (400 MHz, CHCl$_3$, 297 K): $\delta$ 8.28 (s, 12H, Ar-H), 5.79 (t, $J$ = 6.1 Hz, 2H, NH), 4.58 (t, $J$ = 4.6 Hz, 2H, NH), 4.02 (s, 36H, OCH$_3$), 3.26-3.10 (m, 4H, CH$_2$), 2.12-2.06 (br, 6H, C$_{10}$H$_{15}$), 1.85-1.66 (br, 24H, C$_{10}$H$_{15}$), 0.65-0.51 (br, 4H, CH$_2$), −0.39 - −0.50 (br, 4H, CH$_2$).

HRMS (MALDI-TOF, DCTB; Ionizer, NaTFA): calcd. for C$_{78}$H$_{98}$O$_4$N$_4$O$_{14}$: 1314.7074, found: $m/z$ 1314.7074 ([M]$^+$).

### Chemical oxidation of [2]rotaxane (2)

To a CDCl$_3$ solution (0.7 mL) of [2]rotaxane (**2**) (2.1 mg, 1.6 μmol) was added Magic Blue (2.9 mg, 3.5 μmol, 2 equiv.) under an argon atmosphere. The $^1$H NMR spectrum of this sample was measured (Fig. 7).

## Data availability

The crystallographic data (CIF files) for the structures reported in this Article have been deposited with the Cambridge Crystallographic Data Centre (CCDC), under deposition numbers CCDC 2222648 (**Au-1**), CCDC 2222611 (**1**), CCDC 2222615 (**1²⁺**) and CCDC 2222612 (**1⊃G1**). These data can be obtained free of charge from CCDC at [www.ccdc.

cam.ac.uk/data_request/cif]. All Cartesian coordinates of the optimized geometries, relaxed scans, and NEB calculations are provided in a separate file in XYZ format (Supplementary Data 1). The authors declare that all other data supporting the findings of this study are available within the article and its Supplementary Information file. All experimental data are available from the corresponding authors upon request. Source data are provided with this paper.

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

## Acknowledgements

This work was financially supported by a Grant-in-Aid for Young Scientists (JSPS KAKENHI grant number JP19K15533 for Y.T.), Grants-in-Aid for Scientific Research (C) (JSPS KAKENHI grant number 21K05093 for K.O.), Grants-in-Aid for Transformative Research Area (A) "*Condensed Conjugation*" (JSPS KAKENHI grant numbers 21H05496 for Y.T. and 23H04041 for Y.T.). The computation was partly performed using Research Center for Computational Science, Okazaki, Japan (project numbers 21-IMS-C158 for T.I. and 22-IMS-C158 for T.I.). This article is dedicated to Professor Shigeru Yamago on the occasion of his 60th birthday.

## Author contributions

Y.T. and H.K. conceived the project. N.N. synthesized the CPP derivative and examined its properties regarding guest inclusion and chemical oxidations. Y.K. accomplished the synthesis and chemical oxidation of [2]rotaxane. T.I. conducted all the theoretical calculations. H.K. and Y.T. carried out X-ray crystallographic measurements and analysis. Y.T. drafted the initial manuscript, while H.K. and K.O. subsequently revised it.

## Competing interests

The authors declare no competing interests.
