## [Peer review file · Nature Communications]

REVIEWER COMMENTS

Reviewer #1 (Remarks to the Author):

The manuscript by Tsuchido, Kawai, Ide and co-workers describes the gold-templated synthesis of [6]cycloparaphenylene with 12 methoxy groups attached ([6]CPP-12OMe) as well as its redox properties and host-guest chemistry. The experiments as well as theoretical calculations have been executed extensively and properly.

CPPs have now become famous and hot nanohoop molecules and there are plenty of CPP derivatives reported and available. There are several synthetic methods for CPPs but the authors' group reported a very unique and highly original way of constructing CPP frameworks via a macrocyclic gold complex in 2020 (ref 15, 16). In this work, the authors have succeeded in synthesizing densely methoxylated CPP, [6]CPP-12OMe, by their gold-templated method. Unlike other CPP synthetic methods, the gold-templated method has an advantage over sterically congested aryl-aryl bond formation, and the present work nicely demonstrates the advantage.

From the molecular structure point of view, the successful synthesis of highly symmetric and densely oxygenated CPP is a welcome addition to CPP library, but in my eyes the novelty of [6]CPP-12OMe is not significantly high (particularly for Nat. Commun.). As mentioned by the authors (Fig. 1), multiply alkoxyated CPPs have been already reported by Gaeta, Tanaka and Du. Moreover, multiply (and even fully) substituted CPPs have already been reported by other groups. The authors try to emphasize the paucity of such examples limiting to small [5]- and [6]CPP analogues, but this is not a very strong appeal.

Twelve methoxy groups on [6]CPP framework can potentially have a huge and/or surprising impact on the properties including absorption/emission, redox, and host-guest chemistry, all of which have been already extensively explored in CPPs or oxygenated CPPs. The rise of HOMO/LUMO energy levels and the ease of oxidation of [6]CPP-12OMe is an expected trend and also in line with the previous studies on alkoxyated CPPs. Chemical oxidation of [6]CPP-12OMe resulting in the formation of its cation radical and dication is somewhat similar to the report by Yamago for [8]CPP oxidation chemistry (including the emergence of in-plane aromaticity). While the host-guest chemistry of CPP derivatives have become somewhat the "must" execution in CPP community (and Gaeta reported the host-guest chemistry of [8]CPP-2OMe with pyridinium cations), the present findings to include linear alkyl chain (in a gauche conformation) is interesting. In my opinion, perhaps this part is most intriguing result in this paper. The authors went on in this direction to further find that the formation of a [2]rotaxane with [6]CPP-12OMe is possible. While this looks interesting, Guldi and von Delius have already reported CPP-based [2]rotaxanes in 2018 (ref 41). The authors also demonstrated that the dication of their [2]rotaxane is also possible by chemical oxidation with Magic Blue.

All in all, the present manuscript shows the power of their gold-templated CPP synthesis and that [6]CPP-12OMe is interesting in terms of its structure, photophysical properties, redox properties, and host-guest properties. In addition, I should emphasize that the work has been done properly in very high standard. Therefore, this paper should be published in the present form (no complaints whatsoever). However, to be honest, I feel that it does not clear the bar of Nat. Commun. In my eyes, it's suitable for JACS or Angew Chem.

Reviewer #2 (Remarks to the Author):

This manuscript by Narita et al presents the synthesis of a [6]cycloparaphenylene (CPP) with two methoxy groups on all six para-phenylene units. The presence of these methoxy groups makes the macrocycle easier to oxidize, and makes it able to bind guests such as adiponitrile and hexamethylene diisocyanate in chloroform. The authors exploited this host-guest chemistry to prepare a rotaxane.

The important point about this rotaxane is that it enables the magnetic shielding at the center of the dication of the CPP to be measured by NMR; a change in chemical shift of about 15 ppm is recorded. This is the first clear demonstration that oxidized CPP dications display global aromatic ring currents.

The work has been carried out to a very high standard and all the compounds are well characterized. The results are exciting and will attract great interest. This is the first demonstration of this type of CPP host-guest chemistry. I have no hesitation in recommending that the manuscript should be accepted for publication in *Nat. Commun.* after a few minor revisions.

The authors say on page 7 of the manuscript that the presence of the 12 OMe groups increases the ring strain in the [6]CPP, which seem intuitively reasonable, but in the SI (page 21) they say that DFT calculations indicate that the presence of the 12 OMe groups reduces the ring strain. This inconsistency needs to be addressed and the DFT calculations need to be mentioned in the main text.

The authors mention Anderson's use of hexa(4-pyridylethynyl)benzene to probe ring currents in porphyrin nanorings and cite ref 43 [*Nature* 541, 200–203 (2017)], but that template is not mentioned in this reference; they should cite *Nat. Chem.* 12, 236–241 (2020).

When the authors discuss the two stereoisomers on page 6, it would be clearer to use the word "enantiomers".

Reviewer #3 (Remarks to the Author):

This is an excellent article! The design and synthesis of functionalized carbon nanohoops is of particular importance to access highly aesthetic molecular constructs with potential applications in electronics and catalysis. This communication by Tsuchido and coworkers describes an efficient synthetic route to a [6]CPP derivative containing 2,5-dimethoxy-benzene subunits, relying on their well-established Au-templated synthesis. This is a significant addition to the burgeoning field of carbon nanohoop materials. Significantly, the newly prepared CPP molecules are capable of encapsulating linear bisnitrile, which allows for the preparation of rotaxane upon end capping. This information will provide useful guidance for researchers in this field to develop more complex mechanically interlocked molecules based on carbon nanohoops. Overall, this work is scientifically sound, and reflects a large amount of work on organic synthesis, interpreting molecular structures, conducting related characterizations, as well as analysis of data. It does appear to appeal to broad interest of the fields of organic chemistry, coordination chemistry, and supramolecular chemistry. The manuscript is also well written, and the reviewer enjoy reading it. Therefore, the reviewer strongly recommends its publication in *Nature Communications* as it is.

Reviewer #1:

The manuscript by Tsuchido, Kawai, Ide and co-workers describes the gold-templated synthesis of [6]cycloparaphenylene with 12 methoxy groups attached ([6]CPP-12OMe) as well as its redox properties and host-guest chemistry. The experiments as well as theoretical calculations have been executed extensively and properly.

CPPs have now become famous and hot nanohoop molecules and there are plenty of CPP derivatives reported and available. There are several synthetic methods for CPPs but the authors' group reported a very unique and highly original way of constructing CPP frameworks via a macrocyclic gold complex in 2020 (ref 15, 16). In this work, the authors have succeeded in synthesizing densely methoxylated CPP, [6]CPP-12OMe, by their gold-templated method. Unlike other CPP synthetic methods, the gold-templated method has an advantage over sterically congested aryl-aryl bond formation, and the present work nicely demonstrates the advantage.

From the molecular structure point of view, the successful synthesis of highly symmetric and densely oxygenated CPP is a welcome addition to CPP library, but in my eyes the novelty of [6]CPP-12OMe is not significantly high (particularly for Nat. Commun.). As mentioned by the authors (Fig. 1), multiply alkoxyated CPPs have been already reported by Gaeta, Tanaka and Du. Moreover, multiply (and even fully) substituted CPPs have already been reported by other groups. The authors try to emphasize the paucity of such examples limiting to small [5]- and [6]CPP analogues, but this is not a very strong appeal.

Twelve methoxy groups on [6]CPP framework can potentially have a huge and/or surprising impact on the properties including absorption/emission, redox, and host-guest chemistry, all of which have been already extensively explored in CPPs or oxygenated CPPs. The rise of HOMO/LUMO energy levels and the ease of oxidation of [6]CPP-12OMe is an expected trend and also in line with the previous studies on alkoxyated CPPs. Chemical oxidation of [6]CPP-12OMe resulting in the formation of its cation radical and dication is somewhat similar to the report by Yamago for [8]CPP oxidation chemistry (including the emergence of in-plane aromaticity). While the host-guest chemistry of CPP derivatives have become somewhat the "must" execution in CPP community (and Gaeta reported the host-guest chemistry of [8]CPP-2OMe with pyridinium cations), the present findings to include linear alkyl chain (in a gauche conformation) is interesting. In my opinion, perhaps this part is most intriguing result in this paper. The authors went on in this direction to further find that the formation of a [2]rotaxane with [6]CPP-12OMe is possible. While this looks interesting, Guldi and von Delius have already reported CPP-based [2]rotaxanes in 2018 (ref 41). The authors also demonstrated that the dication of their [2]rotaxane is also possible by chemical oxidation with Magic Blue.

All in all, the present manuscript shows the power of their gold-templated CPP synthesis and that [6]CPP-12OMe is interesting in terms of its structure, photophysical properties, redox properties, and host-guest properties. In addition, I should emphasize that the work has been done properly in very high standard. Therefore, this paper should be published in the present form (no complaints whatsoever). However, to be honest, I feel that it does not clear the bar of Nat. Commun. In my eyes, it's suitable for JACS or Angew Chem.

Response: We sincerely appreciate your advice and suggestions concerning the novelty and challenges in the field of CPP chemistry. To introduce the multiple functional groups into a small CPP with a highly distorted structure remains very challenging through the conventional CPP synthesis methods, despite the promising potential for significant modulation in their physical properties. In this study, we achieved the synthesis of [6]CPP with twelve methoxy groups using our originally developed Au-templated method, unveiling distinctive properties arising from its expansively curved cyclic π -conjugated structure, including enhanced oxidation properties, guest inclusion ability for alkyl chained molecules, and the in-plane aromaticity unveiled through a rotaxane structure. We believe these findings are profoundly innovative and of great interest to readers of the *Nature Communications*.

Reviewer #2:

This manuscript by Narita et al presents the synthesis of a [6]cycloparaphenylene (CPP) with two methoxy groups on all six para-phenylene units. The presence of these methoxy groups makes the macrocycle easier to oxidize, and makes it able to bind guests such as adiponitrile and hexamethylene diisocyanate in chloroform. The authors exploited this host-guest chemistry to prepare a rotaxane. The important point about this rotaxane is that it enables the magnetic shielding at the center of the dication of the CPP to be measured by NMR; a change in chemical shift of about 15 ppm is recorded. This is the first clear demonstration that oxidized CPP dications display global aromatic ring currents.

The work has been carried out to a very high standard and all the compounds are well characterized. The results are exciting and will attract great interest. This is the first demonstration of this type of CPP host-guest chemistry. I have no hesitation in recommending that the manuscript should be accepted for publication in Nat. Commun. after a few minor revisions.

Response: We sincerely appreciate the reviewer's evaluations of our manuscript and the important suggestions. We have implemented the suggested revisions based on the comments from the reviewers.

The authors say on page 7 of the manuscript that the presence of the 12 OMe groups increases the ring strain in the [6]CPP, which seem intuitively reasonable, but in the SI (page 21) they say that DFT calculations indicate that the presence of the 12 OMe groups reduces the ring strain. This inconsistency needs to be addressed and the DFT calculations need to be mentioned in the main text.

Response: Thank you for the suggestion. The description in the original manuscript was unclear. We conclude that intramolecular CH...O hydrogen bonds between methoxy oxygens and aryl hydrogens of another benzene ring would be responsible for the smaller strain energy of compound **1** than [6]CPP. To assist in the above discussion, we conducted the additional computational studies (BCPs/QTAIM and NCI). The description that was a problem in the original manuscript (page 6, L18-20) has been removed, and a new sentence has been added as follows. In addition, the results of the computational studies have been added to Figs. S17-18 in the SI.

Additional text (page 6): The strain energy of **1**, as simulated through the DFT (*Density Functional Theory*) calculations (ω B97M-V/def2-TZVP//TPSS-D3(BJ)/def2-SVP level of theory) based on the homodesmotic reaction using ORCA^[34] (see Supplementary Table S3), was unexpectedly estimated to be lower (88.2 kcal mol⁻¹) compared to that of [6]CPP (92.6 kcal mol⁻¹) calculated at the same level, despite the introduction of bulky methoxy groups. Further computational studies, including BCPs/QTAIM (*Bond Critical Points/Quantum Theory of Atoms In Molecules*) and NCI (*Non-Covalent Interaction*) analyses using Multiwfn^[35] and NCIPlot^[36] programs, have demonstrated the formation of intramolecular CH...O hydrogen bonds between methoxy oxygens and aryl hydrogens of another benzene ring (see Supplementary Figs. S17-18). Thus, the lower strain energy of **1** compared to [6]CPP can be attributed to a contribution from stabilization through the multiple hydrogen bonds. This interaction tends to be weak in a less-strained molecule such as (C₆H₄-2,5-OMe)₃H₂^[32] by a large Ar-Ar dihedral angle primarily due to the steric hindrance. Therefore, the formation of the tubular conformation in **1** can be ascribed to the restricted change in dihedral angles due to the strained macrocyclic structure, along with the structure-specific intramolecular CH...O hydrogen bonds.

Additional references (page 17):

[35] Lu, T. & Chen, F. Multiwfn: A multifunctional wavefunction analyzer. *J. Comput. Chem.* **33**, 580–592 (2012).

[36] Johnson, E. R. *et al.* Revealing Noncovalent Interactions. *J. Am. Chem. Soc.* **132**, 6498–6506 (2010).

Additional Figures (Figs S17-18 in the SI):

Figure S17. BCPs (*Bond critical points*) and bond path analysis based on QTAIM (*Quantum theory of atoms in molecules*) of (a) [6]CPP and (b) [6]CPP-12OMe (**1**) (Wavefunction: ORCA 4.2.1, TPSS-D3(BJ)/def2-SVP, QTAIM: Multiwfn). The bond paths regarding the hydrogen bonds and bond critical points (gray lines and orange dots, respectively) are found between aryl hydrogens and methoxy oxygens in **1**, indicating the existence of hydrogen bonds.

Figure S18. NCI (*Non-Covalent Interaction*) plots of (a) [6]CPP and (b) [6]CPP-12OMe (**1**) (Wavefunction: ORCA 4.2.1, TPSS-D3(BJ)/def2-SVP, NCI: NCIPLOT).

The authors mention Anderson's use of hexa(4-pyridylethynyl)benzene to probe ring currents in porphyrin nanorings and cite ref 43 [Nature 541, 200-203 (2017)], but that template is not mentioned in this reference; they should cite Nat. Chem. 12, 236-241 (2020).

Response: We added the suggested paper as reference 46 at the revised manuscript.

When the authors discuss the two stereoisomers on page 6, it would be clearer to use the word "enantiomers".

Response: Following the reviewer's comment, we changed the word "stereoisomers" in the manuscript to "enantiomers".

Reviewer #3:

This is an excellent article! The design and synthesis of functionalized carbon nanohoops is of particular importance to access highly aesthetic molecular constructs with potential applications in electronics and catalysis. This communication by Tsuchido and coworkers describes an efficient synthetic route to a [6]CPP derivative containing 2,5-dimethoxy-benzene subunits, relying on their well-established Au-templated synthesis. This is a significant addition to the burgeoning field of carbon nanohoop materials. Significantly, the newly prepared CPP molecules are capable of encapsulating linear bisnitrile, which allows for the preparation of rotaxane upon end capping. This information will provide useful guidance for researchers in this field to develop more complex mechanically interlocked molecules based on carbon nanohoops. Overall, this work is scientifically sound, and reflects a large amount of work on organic synthesis, interpreting molecular structures, conducting related characterizations, as well as analysis of data. It does appear to appeal to broad interest of the fields of organic chemistry, coordination chemistry, and supramolecular chemistry. The manuscript is also well written, and the reviewer enjoy reading it. Therefore, the reviewer strongly recommends its publication in Nature Communications as it is.

Response: We sincerely appreciate the reviewer's evaluations of our manuscript. Your favorable feedback serves as motivation for us to advance our research further.

REVIEWERS' COMMENTS

Reviewer #2 (Remarks to the Author):

This manuscript has been revised to fully address the referees' comments. It presents an exciting and original piece of work. The manuscript is now suitable for publication without change.

Reviewer #2:

This manuscript has been revised to fully address the referees' comments. It presents an exciting and original piece of work. The manuscript is now suitable for publication without change.

Response: We would like to thank again the reviewer sincerely for your evaluation of the manuscript and for your important comments.